# Scale-Dependent Impacts of Urban Morphology on Commercial Distribution: A Case Study of Xi'an, China



**Fan Liang** [1,2], **Jianhong Liu** [1,*], **Mingxing Liu** [1], **Jingchao Zeng** [1], **Liu Yang** [1] **and Jianxiong He** [1,3]

[1] Shaanxi Key Laboratory of Earth Surface System and Environmental Carrying Capacity, College of Urban and Environment Sciences, Northwest University, Xi'an 710127, China; 20B934025@stu.hit.edu.cn (F.L.); mingxingliu@stumail.nwu.edu.cn (M.L.); zengjingchao@stumail.nwu.edu.cn (J.Z.); 20175193@nwu.edu.cn (L.Y.); jxhe@nwu.edu.cn (J.H.)

[2] College of Architecture, Harbin University of Technology, Harbin 150006, China

[3] Xi'an Northwest University Research Institute of Urban-Rural Planning and Environmental Engineering Co., Ltd., Xi'an 710069, China

**\*** Correspondence: jhliu@nwu.edu.cn

**Abstract:** How to create a sustainable urban morphology for the development of cities has been an enduring question in urban research. Therefore, quantitatively measuring the current relationship between urban morphology and urban function distribution is the key step before urban planning practice. However, existing studies only examine the relationship at limited scales or with a single unit. To comprehensively understand the relationship between urban morphology and commercial distribution, this study utilized space syntax and point of interest (POI) data (shopping and food service) and took the city of Xi'an, China as a case study. The evaluation of relationships was performed with two measurement units (500 m × 500 m grids and street blocks) at 16 different scales (from $R = 800$ m to $R = n$) by engaging three statistical metrics (mean, maximum, and total). Great variations in the relationships between urban morphology and commercial distribution across scales were observed in the study area at both grid level and block level. However, the change trends of the correlation across scales differ substantially when measured by grids and blocks. Generally, the correlations measured by blocks were stronger than those measured by grids, indicating it is desirable to perform such research at the block level. The correlations were stronger at the small scales ($R = 800$ m to $R = 3600$ m) when measured with grids, and the stronger correlations were detected at large scales ($R = 5$ km to $R = 35$ km) when measured with blocks. The strongest correlations were found at the scale $R = 3600$ m with grid unit, and the strongest correlations were detected at the scale $R = 10$ km with blocks. Among the three space syntax variables, urban morphology measured by integration presents stronger correlation with commercial distribution than choice and complex variable for both shopping and food services. This reveals that the centrality of urban space has a greater impact on the locations of commercial establishments than accessibility and comprehensive potential. As for the three statistical metrics, the total is less useful in measuring the impacts of urban morphology on commercial distribution across scales. However, regardless of measurement by grids or by blocks, urban morphology has a stronger impact on the locations of shopping businesses than on food shops. Based on our findings, it is preferable to predict the potential commerce locations by measuring the centrality of the study area at a scale of 10–20 km. Our method can be easily transferred to other urban regions, and the derived results can serve as a valuable reference for government administrators or urban planners in allocating new commerce establishments.

**Keywords:** space syntax; POI; urban morphology; commercial distribution; quantitative analysis; correlation analysis; aggregate method





## 1. Introduction

In the past several decades, urban space sprawl and population migration from rural areas to cities have resulted in rapid global urbanization [1,2]. As a consequence,

urbanization and population expansion bring tremendous challenges to cities, such as severe ecological problems [3], conflicts between various functions of urban space [4], and urban function mismatch [5]. Therefore, current urban planning has become increasingly focused on developing a rational urban spatial structure and the optimal allocation of urban land use and activities [6] to ensure the sustainability and livability of cities, especially for developing countries [7,8].

Urban space structure is a multi-scale concept that is determined by both morphological and functional terms, to describe the spatial configuration and linkages between different activity nodes in the city [9–11]. Generally speaking, the spatial attributes (e.g., space accessibility and space utility) and participants both affect the development orientation of cities [6,12]. Urbanization greatly reshapes urban morphology [12], and economic development at different stages has also altered the distribution of urban functions. The synergic evolution of urban spatial structure and function will promote the sustainable development of a city [3]. However, the asynergic evolution will bring negative impacts. Therefore, accurate quantification of the relationship between urban spatial structure and urban function distributions is a crucial step before urban planning practice.

Space syntax is a spatial fabric analysis method proposed by Bill Hillier and Julienne Hansen [7]. It simulates cities by generating a street topology model and zooming in on a variety of behavior patterns [13]. Space syntax provides a rational and scientific way to analyze urban space [14]. Recently, the ready availability of tremendous digital traces of urban activities has substantially powered quantitative urban study [15]. Great swathes of information are collected by the mass mining, processing, and applications of big data [16]. Among such approaches, point of interest (POI) data are notable because they are collected from multiple sites and convey geographic information. Therefore, POI data can clearly reflect the functional distribution of urban space and thus facilitate urban research [17]. The mathematical model can quantitatively analyze the attributes of urban morphology to improve the objectivity and scientificalness of research [18,19], and adding big data could provide much more urban information, while also reducing the cost and energy [20,21].

At present, some studies employ space syntax and big data (especially POI data) to evaluate the quality of urban streets or the relationship between streets and their surrounding urban functions [22–25]. However, these studies mainly use straight lines (i.e., streets), and no relevant analysis of urban land is involved. Though a few studies have utilized polygon data, they only used grids as the measurement units [26]. However, this ignores the texture of urban space. Moreover, usually only one single measurement metric (i.e., mean value) is used in this kind of study [5,16,27]. Considering the complexity of urban spatial structure, it is not sufficient to fully understand the relationships between urban morphology and urban function distribution.

To bridge the above research gap, the main purpose of this research is to investigate the relationships between urban morphology and commercial distribution at different scales with two measurement units (500 m × 500 m grids vs. urban street blocks). We consider three space syntax variables to measure the street characteristics within each measurement unit. To comprehensively understand the relationship between urban morphology and commercial distribution, we also use three statistical metrics (i.e., the mean value, the maximum value, and the total value). Specifically, we are committed to answering the following questions:

(1) Is the relationship between urban morphology and commercial distribution based on blocks stronger than that based on grids?
(2) How does the relationship change at different scales? At which scale is the relationship the strongest?
(3) Is it possible to predict the urban function distribution characteristics of urban land from the perspective of urban morphology?

## 2. Methodology

### 2.1. Study Area

This study focuses on the scale-dependent impacts of urban morphology on commercial distribution; thus, we take the urban area Xi'an (Figure 1) as an example. Xi'an is the capital city of Shaanxi Province, China. It is located between 107°40′–109°49′ E and 33°42′–34°45′ N. Compared with other cities, Xi'an is most representative of the existing ancient Chinese urban civilization with complex land use and diverse urban structure, as it are the ancient capitals of thirteen dynasties and the starting point of the Silk Road [28,29]. Due to the long history and urban planning concepts of different dynasties, the urban morphology of Xi'an is complex and diverse, including mixed land use and regular street types. In addition, Xi'an is also the most important central city in Northwest China [30]. According to Xi'an Municipal Bureau of Statistics, the population of Xi'an has reached 10.2 million in 2019 and the urbanization level was 74.6% at that time [31]. Since 1978, Xi'an has experienced rapid urbanization growth, resulting in many urban problems, such as urban sprawl, population surge, urban heat, and the deterioration of the urban environment [32,33]. Under the conflicts between historical preservation and modern urban development, its urban morphology has become more complicated, especially the urban functional structure.

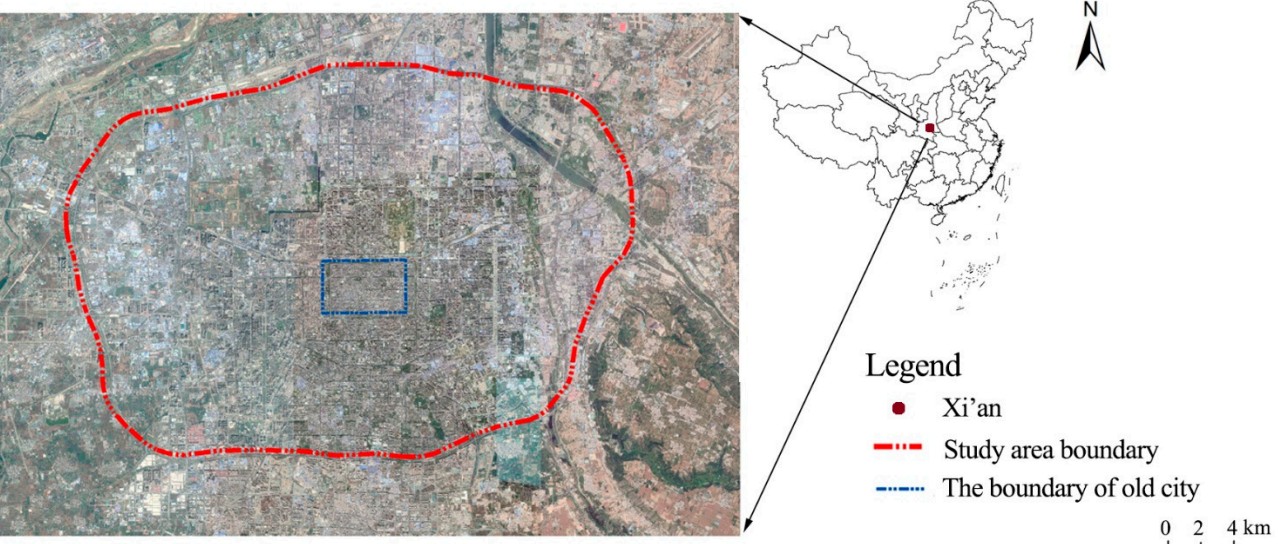

**Figure 1.** Location of the study area.

It is worth noting that the research area is not an independent space. In particular, its marginal zone has a certain connection with the space outside the study area. Since space syntax emphasizes the connectivity between urban spaces [34,35], the marginal regions of the study area have an edge effect, which means the external urban morphology will have a certain impact on the edge of the research area. To alleviate the influence of the edge effect on our results, we used the segmentation effect of the expressway on urban space to limit the study area within the Ring Expressway of Xi'an, which covering an area of about 453 km$^2$ (Figure 1).

### 2.2. Data and Processing

To build a quantitative model to study the urban morphology of Xi'an based on space syntax, we extracted 14 different levels of the city's street networks using Baidu Maps' Interceptor [36]. Next, we used Axwoman [34,37] an ArcGIS plug-in to extract the central lines of each road. The extracted central lines were then imported into software called Depthmap [38] to be converted into line segments. Each line segment is the piece of a street

between two street junctions. Finally, we obtained the line segment diagram to build the street networks of the study area (Figure 2).

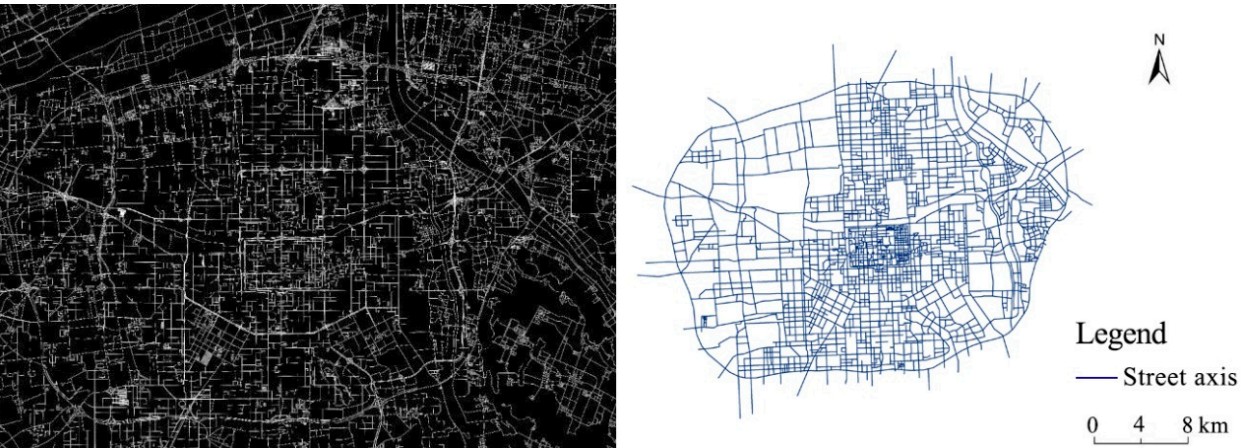

**Figure 2.** The street networks of the study area.

POIs from all commercial sectors are desired for a comprehensive understanding of the relationship between urban morphology and commercial distribution. However, we can only access two kinds of commercial POIs (i.e., food services and shopping) for our study. As food services and shopping are the two basic urban commercial functions [39], they are not only closely related to residents' daily lives but also play a more important part in the commercial functions. The POIs within the research area in 2019 were crawled from Baidu Map (https://map.baidu.com/). The attributes of each POI datum include store name, floor space, geographic location, and other information. We then used the geographic locations of POI data to map the distribution of food service and shopping (Figure 3).

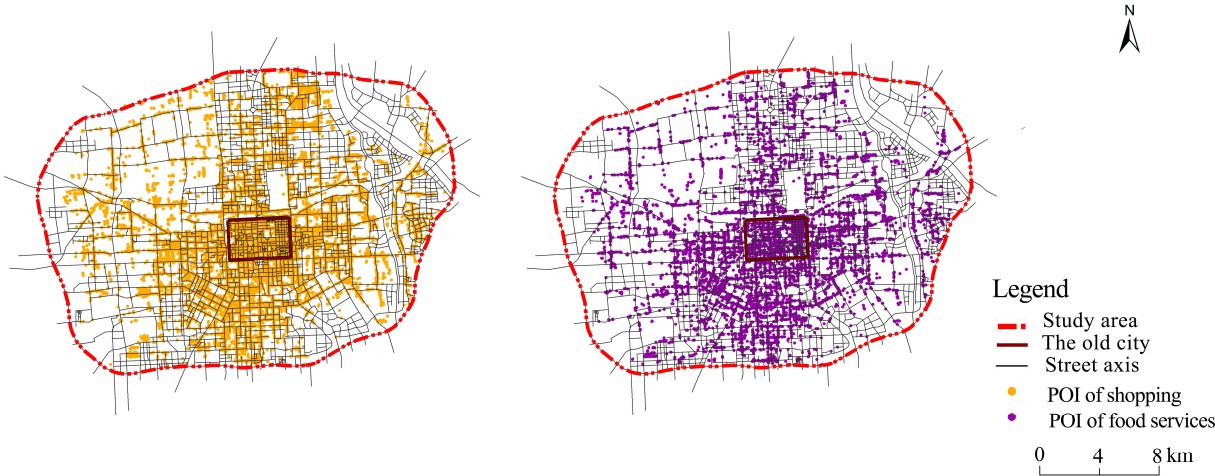

**Figure 3.** The POIs of food services and shopping.

### 2.3. Selection of Variables in Space Syntax

Space syntax focuses on a variety of behavioral patterns and yields a network structure woven by streets. It measures an urban space system by analyzing the topological structure and internal logic of its axes and their combinations to quantify an urban space. At present, the most important and basic variables of space syntax include choice ($C$) [40] and integration ($I$) [41]. Choice embodies space's ability to be traversed [26] and therefore represents the traversal effect of a space. The larger the choice value, the greater the proba-

bility that the space is traversed. In other words, the larger the choice value, the greater the potential of selection. Integration represents the central character of the space and reflects the aggregation effect of the space and other spaces. The larger the integration value, the stronger the centrality of the space. In order to find the potential space, integration and choice can be combined to form a complex variable (*CV*) [41], which indicates the comprehensive value potential of the urban space. This helps to find the segments in a network that serve as both potential destination routes and movement centers [42]. A greater *CV* value signals a higher likelihood of spatial benefits (Table 1).

**Table 1.** Space syntax variables selected in this study.

| Variables | Formula | Parameter | Meaning |
|---|---|---|---|
| Choice | $C_i = \sum\limits_{j}\sum\limits_{k} \frac{d_{jk}(i)}{d_{jk}}(j < k)$ | Where $d_{jk}$ is the shortest path between the $j$-th line and the $k$-th line, $d_{jk}(i)$ is the shortest path including the $i$-th line between the $j$-th line and the $k$th line. | The greater the value, the greater the possibility the space is crossed. |
| Integration | $I_i = \frac{n\left[\log_2\left(\frac{n+2}{3}-1\right)\right]+1}{(n-1)(MD_i-1)}$ | $n$ is the total number of nodes in the connection diagram, $MD_i$ is the average depth value | Greater integration means higher traffic accessibility, stronger centrality, and a more convenient street space. |
| Complex Variable | $CV_i = I_i(\log C_i + 2)$ | $I_t$ is the integration value of the $i$-th part space, $C_i$ is the number of spaces intersecting with the $i$-th part of the space | The larger the complex variable, the greater the potential that the street space will become a spatial activity center. |

### 2.4. Analysis of Relationships at Different Scales with the Two Measurement Units

To investigate the relationship between urban morphology and commercial distribution at each scale, the measurement radius of space syntax provides conditions for studying urban morphology at different scales. Space syntax can be used to conduct multi-scale research by setting different measurement radii, which are the topological distances from a certain urban street to other streets (Figure 4) or the shortest metric distances.

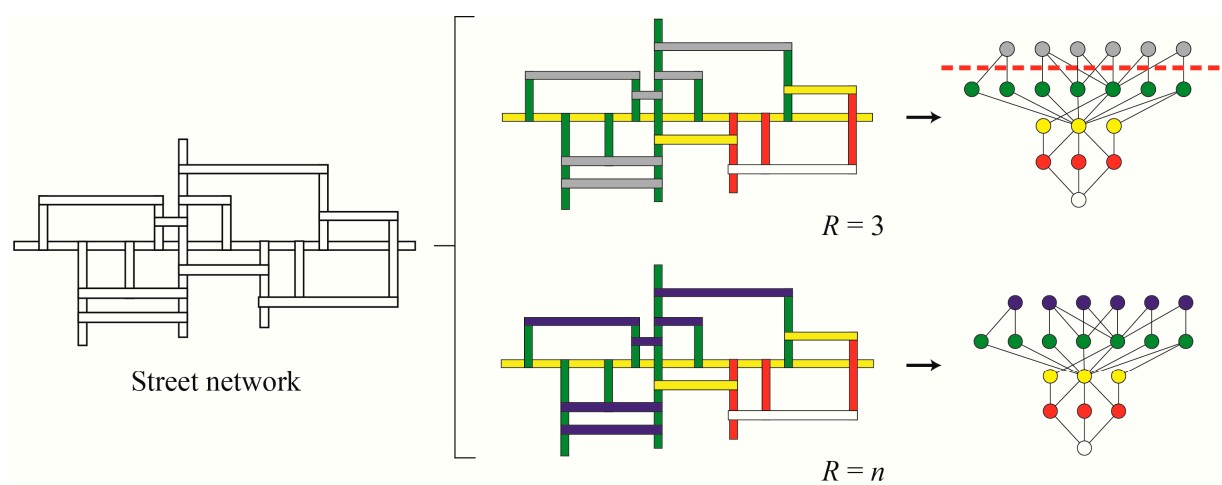

**Figure 4.** The diagrammatic sketch for two topological measurement radii (*R* = 3 and *R* = *n*) of space syntax and the line segments included in the measurement, respectively.

However, urban space with more uniform morphology and structure requires more detailed expression [43], such as Xi'an with the grid-like streets. This expression is the segment expression based on the axis model and road centerline data [44]. Among them, the angle analysis in segment analysis is considered to be very consistent with human behavior [45]. At the same time, the angle analysis under the limit of the metric distance

radius is helpful to extract the main reaching traffic and traversing traffic paths in the urban street network [46]. Therefore, this research is more inclined to use the metric distance radius to establish different research scales. More specifically, the metric distance radius ($R$) refers to the metric distance from each segment along all the available streets and roads from that segment to the radius distance [42]. For example, if the metric radius is 800 m ($R$ = 800 m), then starting at each adjacent segment line, the measure is answerable to an 800 m limit along these lines. Alternatively, if the metric radius is $n$ ($R = n$), then no radius restrictions are put on any city segments. This enables the analysis of the correlations and influencing factors between the urban morphology and the distribution of business under different scales. Accordingly, we first set 16 metric radii from the local scale ($R$ = 800 m) to the global scale ($R = n$) (Table 2). We used DepthMap to analyze the above three space syntax variables 16 times (Figure 5).

**Table 2.** The measurement radii at different scales.

| Scale Class | Radius |
|---|---|
| Small and medium scale | 800 m/1200 m/1600 m/2000 m/2400 m/<br>2800 m/3200 m/3600 m |
| Large scale | 5 km/10 km/15 km/20 km/25 km/30 km/35 km |
| Global scale | $n$ |

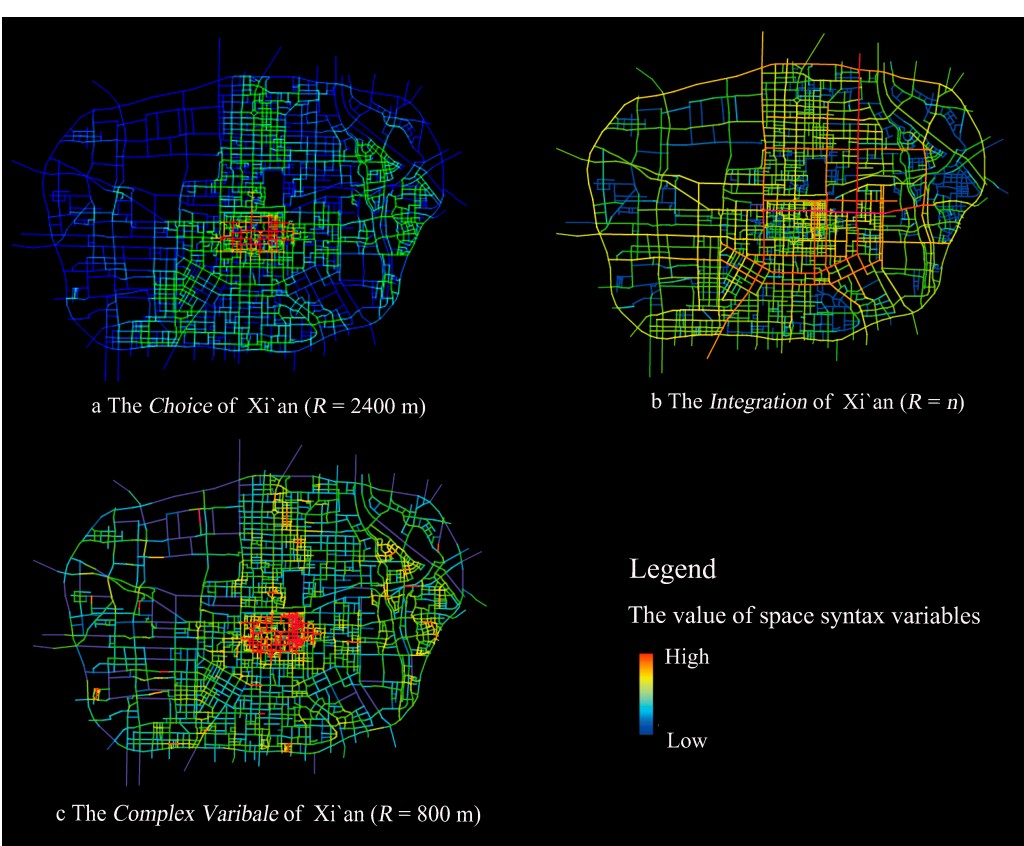

a The *Choice* of Xi`an ($R$ = 2400 m)

b The *Integration* of Xi`an ($R = n$)

c The *Complex Varibale* of Xi`an ($R$ = 800 m)

**Figure 5.** Three examples of space syntax measuring results.

Considering the scale effect and spatial differences of urban space, this study first used a 500 m × 500 m grid to uniformly divide the urban space and then used the grid as the measurement unit (Figure 6). Next, we divided the urban space based on the urban street blocks to characterize the urban texture. Totally, there are 5239 grids and 1559 blocks within the study area.

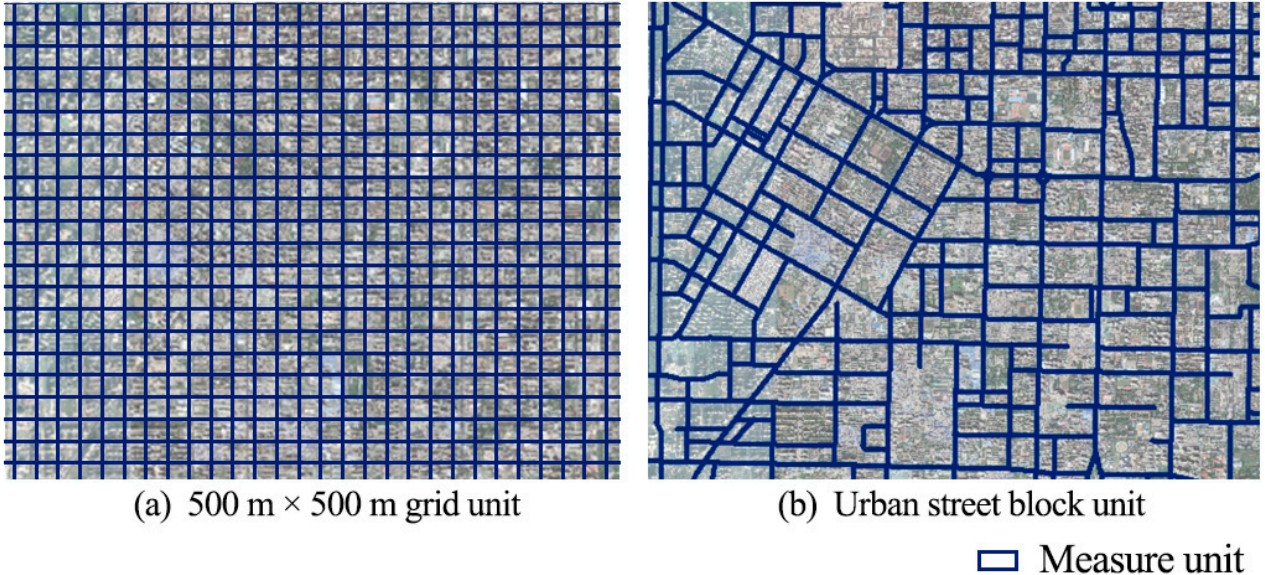

(a) 500 m × 500 m grid unit   (b) Urban street block unit

☐ Measure unit

**Figure 6.** Two types of measurement units considered in this study.

Since the grid size is rather small, the lengths of line segments generally are greater than 500 m; thus, the obtained space syntax variables within a grid are relatively uniform. Therefore, we only calculated the mean value of the obtained space syntax variables within each grid. In contrast, block units are based on the urban texture and show relatively greater heterogeneity in urban morphology between each other. To better illustrate the characteristics of urban morphology within each block unit, we calculated the maximum, average, and total values of the obtained space syntax variables within each block unit (Table 3).

**Table 3.** Commercial distribution measurement metrics and urban morphology measurement metrics with two measurement units.

| Relationship Measurement Unit | Commercial Distribution Measurement Metric | Urban Morphology Measurement Metric |
|---|---|---|
| 500 m × 500 m grid | Log value of the total number of POIs within the measurement unit | The mean value of $C$ <br> The mean value of $I$ <br> The mean value of $CV$ <br> The max value of $C$ <br> The max value of $I$ <br> The max value of $CV$ |
| Urban street block | Log value of the total number of POIs within the measurement unit | The mean value of $C$ <br> The mean value of $I$ <br> The mean value of $CV$ <br> The total value of $C$ <br> The total value of $I$ <br> The total value of $CV$ |

*2.5. Correlation Analysis*

To quantitatively analyze the relationship between urban morphology and commercial distribution in Xi'an, we used Pearson correlation analysis [47,48]. The premise of performing Pearson correlation analysis is that the data exhibit a normal distribution. However, the commercial distribution measurement metrics with grid units and block units both demonstrated non-normal distribution. Hence, we applied the logarithm transformation as used in a previous study before performing the Pearson correlation analysis [26]. The transformation is as Equation (1):

$$X = \log(N+1) \tag{1}$$

where *N* represents the total number of POIs within each measurement unit and *X* represents the value derived after logarithm transformation (i.e., commercial distribution measurement metrics).

After that, we performed the Pearson correlation analysis by using SPSS software to get the correlation coefficients between the calculated urban morphology measurement metrics and the calculated commercial distribution measurement metrics, as described in Table 3. On this basis, a *t*-test was performed to test the significance of the derived correlation coefficients [49]. The correlation coefficient (*r*) was computed as Equation (2):

$$r_{XY} = \frac{Cov(X,Y)}{\sqrt{D(X)}\sqrt{D(Y)}} \tag{2}$$

where *X* represents the commercial distribution measurement metric and *Y* represents the urban morphology measurement metric.

## 3. Results

### 3.1. The Spatial Distributions of Food Service and Shopping

We counted the number of food services and shopping establishments within each 500 m × 500 m grid, as shown in Figure 7a. It is obvious that high-density shopping services are mainly concentrated within the old city of Xi'an and in the south-central area. Moreover, the distribution of shopping services within the old urban area is relatively oversaturated and therefore spills over into the surrounding areas, suggesting that the old urban area is home to the entire district's primary shopping center. Accordingly, the old urban area is a central gathering place and thus bears a lot of pressure. Shopping distribution far away from the old urban area is relatively scattered. In contrast, the high-density of food services is mainly concentrated within the old urban area (Figure 7b). As food services spread outwards, food service density gradually decreases and demonstrates a concentric circle distribution characterized by a relatively obvious aggregation. In addition, areas with food services at medium- and high-density levels are mainly concentrated in the center and south of the study area, with a weaker concentration in the north.

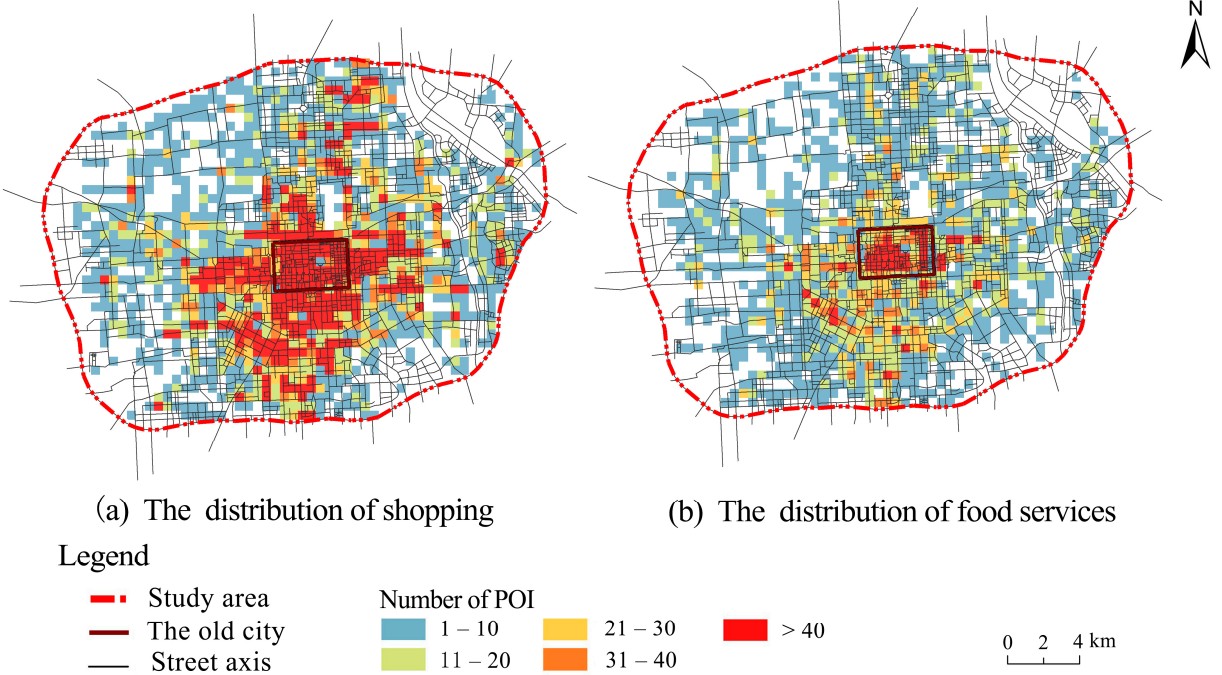

(a) The distribution of shopping   (b) The distribution of food services

Legend

- ▄▄▄ Study area
- ▬▬▬ The old city
- ▬▬▬ Street axis

Number of POI
- 🟦 1 − 10
- 🟩 11 − 20
- 🟨 21 − 30
- 🟧 31 − 40
- 🟥 > 40

0  2  4 km

**Figure 7.** Commercial distribution within the study area measured by grids.

In contrast, we also calculated the densities of food service and shopping within each block. The area of one block is different from that of another; big blocks generally contain more commerce establishments. Therefore, the quantity within each block is divided by its area and the result is denoted by *P'* (Figure 8). Compared with Figure 7, we found that the overall patterns of the commercial distribution were enlarged and the subtle differences were ignored. In addition, the contrast of density becomes less prominent under the block units.

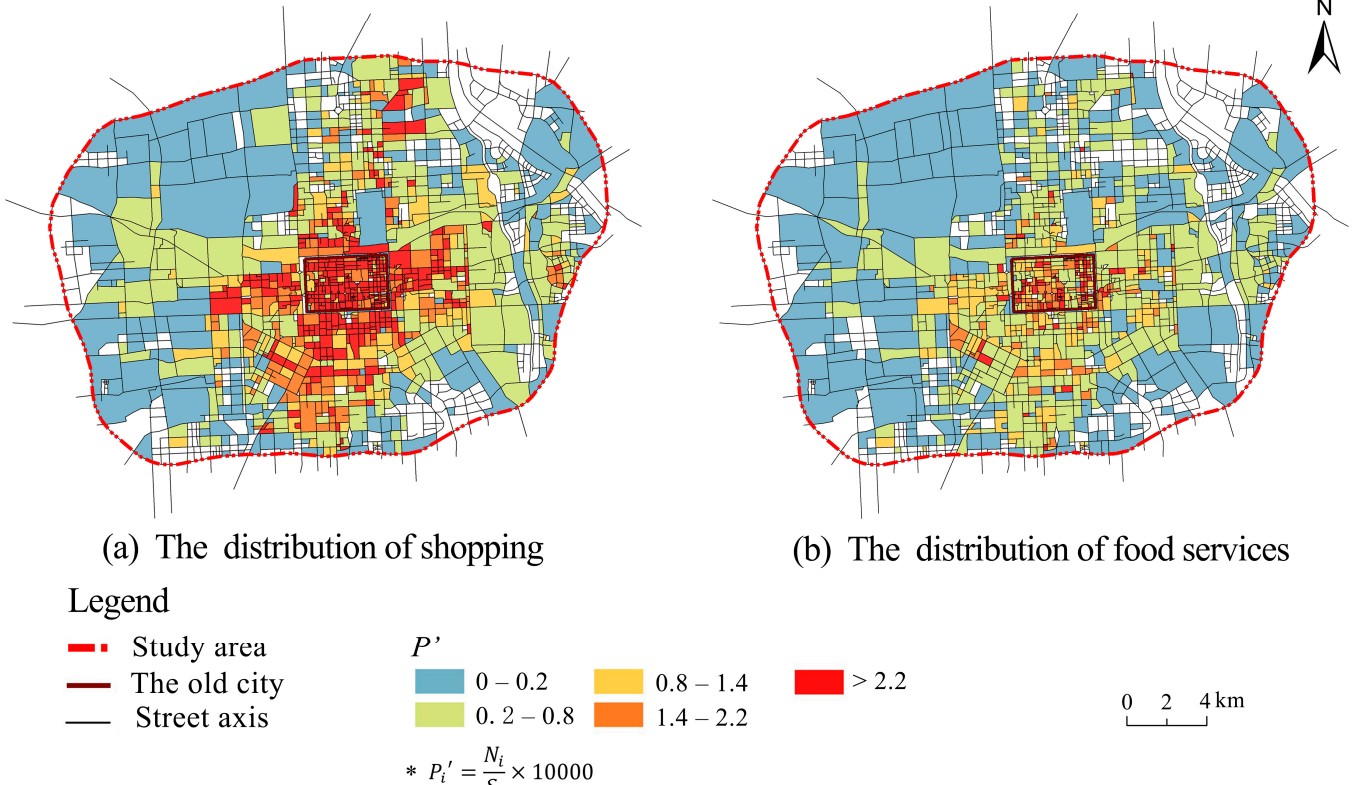

    (a) The distribution of shopping          (b) The distribution of food services

**Legend**

- - · Study area
— The old city
— Street axis

*P'*

| | | |
|---|---|---|
| 🟦 0 – 0.2 | 🟨 0.8 – 1.4 | 🟥 > 2.2 |
| 🟩 0.2 – 0.8 | 🟧 1.4 – 2.2 | |

0  2  4 km

$$* \ P_i' = \frac{N_i}{S_i} \times 10000$$

*Where Ni represents the number of POIs in block i and Si represents the area of block i*

**Figure 8.** Commercial distribution within the study area measured by blocks.

To sum up, food service and shopping distribution under both grid and block units show that the old city of Xi'an is its current commercial center. However, we observed that this old urban area is in a state of oversaturation and suffering from great spatial pressure. There is a trend that the commercial activities are spreading towards the south of the study area. Due to ecological restrictions and the protective development policies, commercial activities are much less active in the north. However, no obvious axis characteristics were found.

*3.2. The Relationship between Urban Morphology and Commercial Distribution Measured by Grids*

After calculating the mean value of each space syntax variable within each grid at 16 scales, we then calculated the urban morphology metrics and commercial distribution metrics within each 500 m X 500 m grid. After that, we calculated the correlation coefficients between urban morphology and commercial distribution. The results are shown in Figure 9. Interestingly, urban morphology measured by integration shows a stronger correlation with commercial distribution than choice and complex variable for both shopping and food services. This reveals that the centrality of urban space has a stronger impact on the location of shopping establishments and food services than accessibility and comprehensive potential. The correlation between integration and commercial distribution first increases

and then decreases when measurement scales become larger. The greatest correlation coefficients were achieved at the radius of 3600 m ($r = 0.416$ for shopping and $r = 0.386$ for food service). Therefore, the centrality of urban space shows a relatively strong impact on commercial distribution within 3600 m, and the impact rapidly declines outside 3600 m.

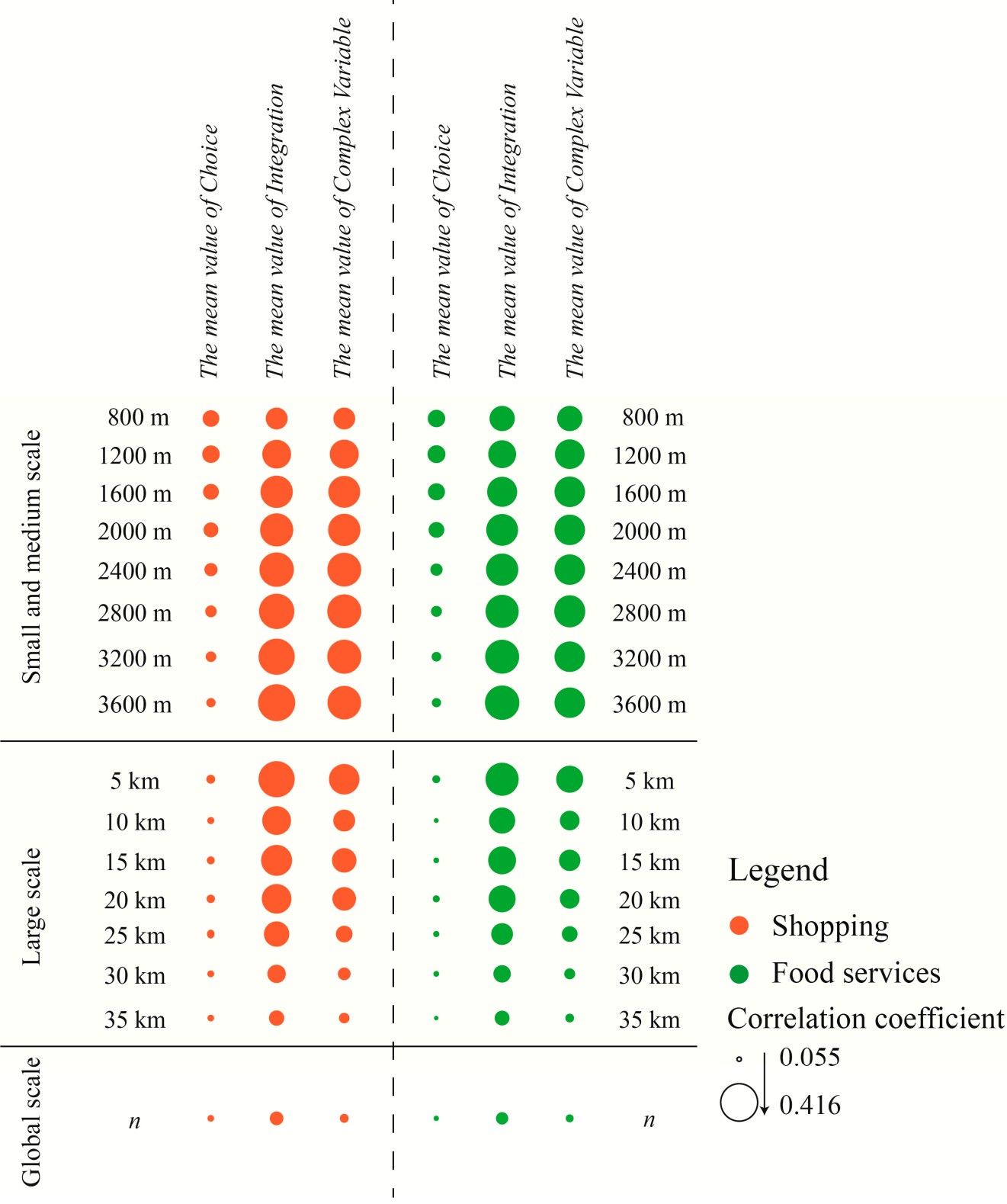

**Figure 9.** The correlation coefficients between urban morphology and commercial distribution measured by grids at different scales (all coefficients are significant at the 0.01 level). N = 5239 for all cases.

In contrast, the impacts of the accessibility (choice) on commercial distribution are very similar to those of centrality, but the magnitude is smaller. Additionally, the strongest impact scale is $R = 2800$ m. However, the comprehensive potential (complex variable) of urban morphology on commercial distribution is relatively low, and the impacts largely are confined within a very small scale ($R = 1200$ m). When treating the entire study area as a whole ($R = n$), the correlation coefficients are all relatively small, which means the radiation ability of a city's centrality, accessibility, and comprehensive potential for commercial distribution is very weak at the global scale.

### 3.3. The Relationship between Urban Morphology and Commercial Distribution Measured by Blocks

Similarly, we calculated the urban morphology metrics and commercial distribution metrics with blocks. Since the sizes of blocks have greatly differed, we used three statistical metrics (mean, maximum, and the total) to compute urban morphology metrics. By performing Pearson correlation analysis, we derived the correlation coefficients between urban morphology and commercial distribution at different scales. As shown in Figures 10 and 11, both shopping and food services show almost the same overall trends under the block unit; only the correlation coefficients of food services are generally smaller than those of shopping establishments. It is easy to note that the change trends of the correlation coefficient are completely different from those presented in Figure 9. When measured at the block level, the relationships between urban morphology and commercial distribution are weak and even negative at small scales and are much stronger at large scales. Besides, the trends presented by the maximum metric and the mean metric of integration, choice, and complex variable are almost the same. The strongest correlation is detected at $R = 10$ km ($r = 0.531$ for shopping and $r = 0.406$ for food service) for integration. Therefore, when measured by blocks, the impact of the centrality of urban morphology on commercial distribution is also stronger than accessibility and comprehensive potential. Moreover, the relationships measured at the block level are generally stronger than relationships detected at the grid level. It can be inferred that it is more preferable to predict the commercial distribution of the study area by measuring the urban special centrality at a scale of 10–20 km. However, the trends showed by the total value are different and the correlation coefficients are smaller. Compared with the mean and maximum metrics, the correlation coefficient change is not very prominent when the urban morphology is measured by the total metric. This means the total metrics are less sensitive to the characteristics of urban morphology at different scales. Therefore, the total metric cannot well capture the differences in urban morphology at different scales. Combining Figures 9–11, it is easy to understand that measurement units greatly affect the research results.

### 3.4. Summary

Figure 12 summarizes the scales of the strongest correlations between commercial distribution and urban morphology measured with space syntax. When measured by 500 m × 500 m grids, the radii of the strongest correlations between commercial distribution and urban morphology derived by integration, complex variable, and choice are 3600, 2800, and 1200 m, respectively. This conclusion can be found for both shopping and food services. The strongest correlation by integration means the potential impact of spatial centrality on commercial distribution is highest. When measured by urban street blocks, the radii of the strongest correlations between commercial distribution and urban morphology differ between shopping and food services. For shopping, the radii with the strongest correlations derived by integration, complex variable, and choice are 10, 15, and 10 km, respectively. For food services, the radii are 10, 20, and 15 km, respectively. Since we used three statistical metrics (i.e., mean, maximum, and total) in the analysis at the block level, the maximum metric shows the strongest correlation. The highest correlation occurred with integration maximum at the scale $R = 10$ km, which means the most central of blocks have the greatest potential to affect commercial distribution at a radius of 10 km. However, whether measured by grids or by blocks, the relationships between urban morphology and

shopping distribution are stronger than those between urban morphology and food service distribution. This reveals that urban morphology has a greater impact on the locations of shopping areas than the locations of food services.

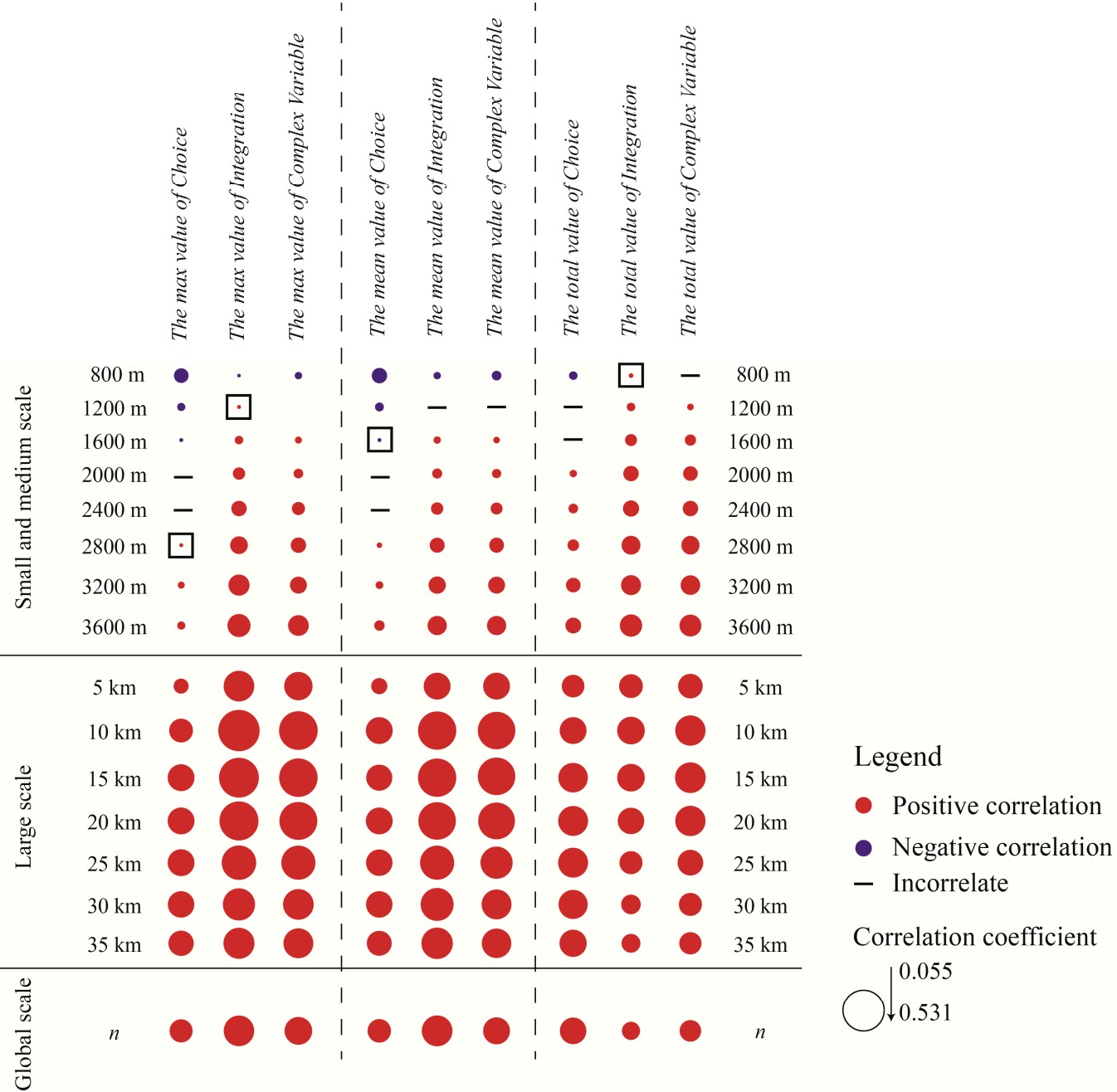

**Figure 10.** The correlation coefficients between shopping distribution and urban morphology measured by blocks at different scales (all coefficients are significant at the 0.01 level except those enclosed by a box, which are significant at the 0.05 level). N = 1559 for all cases.

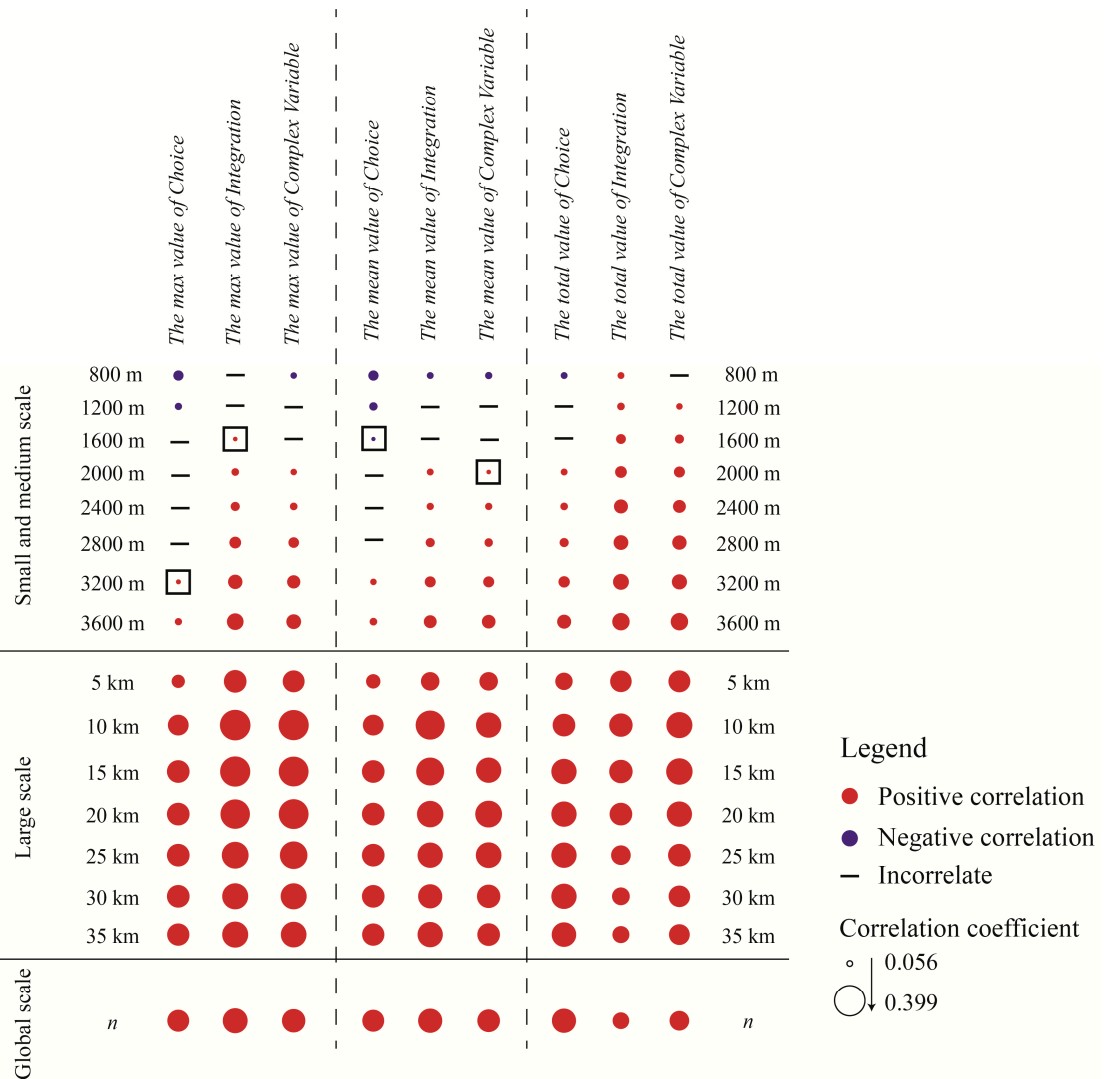

**Figure 11.** The correlation between urban morphology and food services' distribution measured by blocks at different scales. All coefficients are significant at the 0.01 level except those enclosed by boxes, which mean significant at the 0.05 level. N = 1559 for all cases.

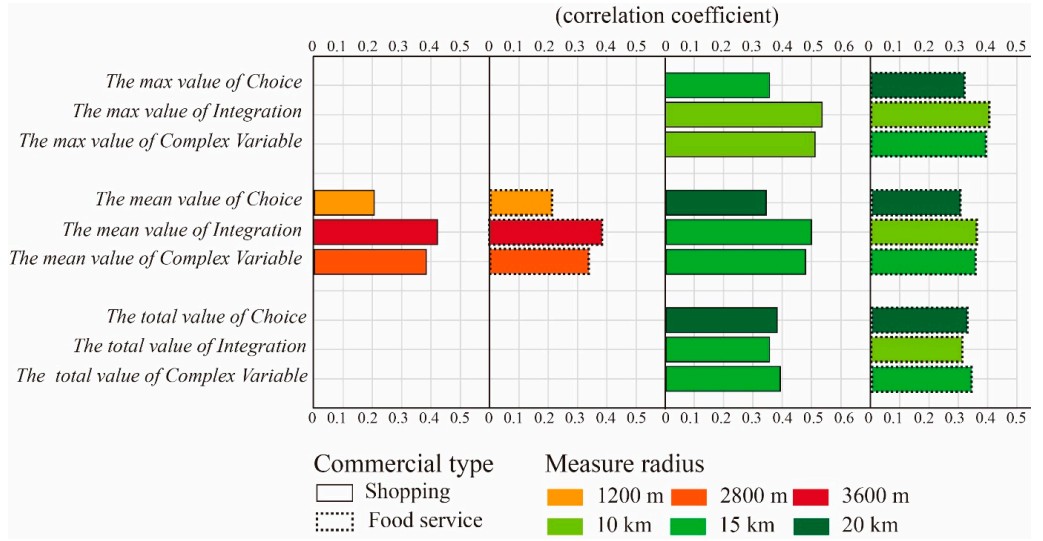

**Figure 12.** The radii of the strongest correlations between commercial distribution and urban morphology measured with space syntax. Orange/red for radii measured by grids and green for radii measured by blocks.

## 4. Discussion

### 4.1. Implications of This Study

How to promote the sustainable development of urban areas by adjusting the urban morphology has been an enduring question in urban research [16]. The quantitative measure of the relationship between urban morphology and commercial distribution is crucial for urban planning practice [50]. However, existing studies only examine the relationship at limited scales or with a single unit [17,26]. To bridge this gap, this study investigated the relationship with grid unit and block unit at 16 scales (from $R = 800$ m to $R = n$) and compared the performances of different statistical metrics (mean, maximum, and total).

Overall, great variations in the relationships between urban morphology and commercial distribution were observed across scales, whether using the grid unit or the block unit. This indicates the impact of urban morphology on commercial distribution is heavily scale-dependent. Therefore, it is easy to understand that measurement units greatly affect the research results. Greater correlation coefficients were achieved with the block unit, which means it is better to use the street blocks to do this kind of analysis. However, the change trends of relationships measured by grids are greatly different from those measured by blocks. The relationships are stronger at the small scales ($R = 800$ m to $R = 3600$ m) when measured with grids, while the stronger relationships are detected at large scales ($R = 5$ km to $R = 35$ km) when measured with blocks. The strongest correlation was found at the scale $R = 3600$ m with the grid unit, but the strongest correlation was detected at the scale $R = 10$ km with the block unit. All experiments show urban morphology measured by integration presents a stronger correlation with commercial distribution than choice and complex variable, which reals the centrality of urban space has greater impacts on the locations of shopping and food services than accessibility and comprehensive potential. As for the three statistical metrics, mean and maximum generally present very similar trends, but the total is less useful in capturing the differences of urban morphology at different scales. Based on our findings, it is preferable to predict the potential commerce locations by measuring the centrality of the study area at a scale of 10–20 km.

In this research, we provide a practical way to evaluate urban land use at multiple scales, especially since the method is based on urban space segments. Research on urban morphology based on the street network has confirmed that urban morphology has a certain influence on urban functions [8]. Since streets are line data, they cannot be directly related to area data such as urban blocks. In response to this phenomenon, we used the grids and urban street blocks. Then street attributes (centrality, etc.) in individual urban space can be summarized by area units. In this way, we can perform analysis on area units. Moreover, space syntax provides us with different research scales; thus, we can conduct a more comprehensive quantitative analysis of urban land use. All of these facilitate new research ideas on urban land evaluation.

Our findings provide new insights into the quantitative analysis of the relationship between urban morphology and commercial distribution. They can serve as a real case reference for researchers and urban planners for choosing the proper unit, scale, metric, and statistic to do this kind of analysis. The method we proposed can be easily applied to other urban regions. It also provides a useful method to predict the potential locations of new commerce under the context of current urban space. The analysis results can serve as a valuable reference for government administrators or urban planners in allocating new commerce establishments.

### 4.2. Potential Development in the Future

It should be noted that, at the block level, urban spatial morphology and commercial distribution have the strongest correlations at a scale of 10–20 km. To understand this, we preliminarily infer that this may be related to the internal connectivity among different parts of urban spaces. Therefore, we simply use another variable, the metric mean depth (*MD*). *MD* can reflect the patchwork maps and can be used to reveal the spatial attributes of

the local dynamic processes of connection, reduction, and diffusion [51]. *MD* is calculated as Equation (3):

$$MD_i = \frac{TD_i}{NC_i} \tag{3}$$

where $MD_i$ is metric mean depth of the *ith* line segment in the study area, $TD_i$ is the cumulative total of the shortest angular paths from the *ith* line segment to all other line segments, and $NC_i$ is the number of line segments encountered on the route from the *ith* line segment to all other line segments.

There are totally 2477 line segments in our study area. We computed the *MD* of each line segment, and then the maximum, minimum, and mean *MD* of all line segments were derived (Table 4). Coincidentally, we found that 10 km and 20 km are close to the mean *MD* and maximum *MD*, respectively. However, whether there is a connection between the two requires further study.

**Table 4.** The maximum, minimum, and mean metric mean depth *(MD)* of the study area.

| Number of Line Segments within the Study Area | Max *MD* | Min *MD* | Mean *MD* |
|---|---|---|---|
| 2477 | 22.132 km | 0.966 km | 12.147 km |

## 5. Conclusions

To comprehensively understand the relationship between urban morphology and commercial distribution, this study investigated the relationship based on space syntax and POI data (shopping and food service). The relationship evaluation was performed with two measurement units (500 m × 500 m grids and street blocks) at 16 different scales (from $R = 800$ m to $R = n$) by engaging three space syntax variables (integration, choice, and complex variable) and three statistical metrics (mean, maximum, and total). Results show that the relationship trends differ substantially when measured by grids and blocks. The strongest correlation was detected at the scale $R = 3600$ m when measured by grids, and the strongest correlation was detected at the scale $R = 10$ km when measured by blocks. Urban morphology measured by integration presents a stronger correlation with commercial distribution than choice and complex variable for both shopping and food services. This reveals that the centrality of urban space has greater potential for impacts on the locations of commercial establishments than accessibility and comprehensive potential. As for the three statistical metrics, the total is less useful in capturing the differences of urban morphology at different scales.

Based on the results of the correlation analysis, the hypothesis for the preliminary prediction of the commercial distribution of Xi'an through measuring the urban spatial centrality at 10 to 20 km is proposed, and further verification is expected. This provides the possibility to realize the preliminary prediction of potential commerce location from the perspective of urban morphology. Our method can be easily transferred to other urban regions, and the derived results can serve as a valuable reference for government administrators or urban planners in allocating new commerce establishments.

However, all conclusions were derived with shopping and food service POIs in our study; whether similar conclusions can be found with other commercial types still needs further verification.

**Author Contributions:** Conceptualization, F.L. and J.L.; methodology, F.L., J.L., and J.H.; software, F.L., M.L., and J.Z.; validation, J.Z. and M.L.; formal analysis, F.L. and J.L.; investigation, F.L. and M.L.; resources, F.L. and L.Y.; data curation, F.L.; writing—original draft preparation, F.L. and J.L.; writing—review and editing, J.L., J.H., and L.Y.; visualization, F.L., M.L. and J.Z.; supervision, J.L.; project administration, J.H.; funding acquisition, J.L. and L.Y. All authors have read and agreed to the published version of the manuscript.

**Funding:** This work was supported by the National Natural Science Foundation of China (number 51908462 and number 41401494).

**Data Availability Statement:** The data presented in this study are available on request from the corresponding author. The data are not publicly available due to the big volume.

**Conflicts of Interest:** The authors declare no conflict of interest.

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
