# Peer review of "Scale-Dependent Impacts of Urban Morphology on Commercial Distribution: A Case Study of Xi’an, China"

_land, doi:10.3390/land10020170_

Round 1
Reviewer 1 Report
The authors used space syntax to reveal the relationship between urban morphology and commercial distribution. The work is timely and interesting. However, there are still a few aspects that should be improved to make the paper publishable. I focus here only on a few major points, which are hopefully easy for the authors to take into account in the revision.
- It is not clear why Xi’an is an important case for research. Why is this case representative? How the lessons learnt from this particular can be extended to the other areas? The authors need to justify the case representativeness to attract a wider audience without local knowledge.
- The literature review has not been well organized. The theoretical foundation of urban activity distributions (and accessibility) is linked to the idea of spatial equilibrium across a city region (i.e., urban spatial structure in urban economics). However, the authors failed to engage with the theoretical work and concepts in their review. Suggest reading, for instance, https://www.jstor.org/stable/2564805 and https://doi.org/10.1016/j.cities.2018.12.015.
- Due to the weak lit review, the theoretical contribution is not clearly summarised in the introduction and conclusions sections. How do the findings specifically contribute to the existing literature, particularly in the research domains of Land?
- Despite many interesting findings, a good construct of the policy implications from the assessment results is needed in the discussion/conclusion. What is the possible take-away for decision-makers and urban planners by understanding the relationship between urban morphology and commercial development?
Author Response
General response:
We greatly thank the four anonymous Reviewers for putting forward the constructive comments essential for improving the quality of our paper. In this version of revision, we substantially revised the Abstract, Introduction, and Discussion by considering all reviewers’ suggestions, and especially addressed the contributions of our study in these parts. We also added some details to the method in order to make it clearer. We reorganized the words of results to better express our findings. We provide two versions of the revised manuscript, one is in track change and one is in no track. For smooth reading, we highly recommend the no track version, as the line number mentioned below are referred to the line position in the no track version. Finally, the whole paper was extensively polished for its language. Below are the specific responses to individual comments.
Response to Reviewer 1
The authors used space syntax to reveal the relationship between urban morphology and commercial distribution. The work is timely and interesting. However, there are still a few aspects that should be improved to make the paper publishable. I focus here only on a few major points, which are hopefully easy for the authors to take into account in the revision.
Q1. It is not clear why Xi’an is an important case for research. Why is this case representative? How the lessons learned from this particular can be extended to the other areas? The authors need to justify the case representativeness to attract a wider audience without local knowledge.
A: It’s good advice. Compared with other cities, Xi'an is most representative of the existing ancient Chinese urban civilization with complex land use and diverse urban structure as it is the ancient capital of thirteen dynasties and the starting point of the Silk Road. In addition, Xi’an is also the most important central city in Northwest China. Under the conflict between historical preservation and modern urban development, its urban morphology has become more complicated, especially the urban functional structure. Therefore, the urban area of Xi’an is a typical example to analyze the scale-dependent impacts of urban morphology on commercial distribution. Please see more details in Section 2.1 in the revised manuscript. Please see the revised manuscript in Line 100-115.
Q2. The literature review has not been well organized. The theoretical foundation of urban activity distributions (and accessibility) is linked to the idea of spatial equilibrium across a city region (i.e., urban spatial structure in urban economics). However, the authors failed to engage with the theoretical work and concepts in their review. Suggest reading, for instance, https://www.jstor.org/stable/2564805 and https://doi.org/10.1016/j.cities.2018.12.015.
A: Thank you for your advice. To better express the objective, as well as to point out the research gap of this study, we have extensively revised the Introduction part, especially increased the basic theoretical work and concepts of urban space structure. Please see Introduction in the revised manuscript. Please see the Abstract in the revised manuscript.
Q3. Due to the weak lit review, the theoretical contribution is not clearly summarized in the introduction and conclusions sections. How do the findings specifically contribute to the existing literature, particularly in the research domains of Land?
A: Thank you for your advice. We greatly rewrote the Introduction and Conclusions in this version. We pointed out the research gap and clarified the theoretical contributions of our paper. Please see the Introduction, Discussion, and Conclusions in the revised manuscript.
Q4. Despite many interesting findings, a good construct of the policy implications from the assessment results is needed in the discussion/conclusion. What is the possible takeaway for decision-makers and urban planners by understanding the relationship between urban morphology and commercial development?
A: It’s good advice. To understand the relationship between urban form and commercial development, decision-makers and urban planners may better formulate urban land use policies and rationally allocate urban functional areas. We have added the policy implications of our assessment results in the Discussion and Conclusion. Please see the corresponding part in the revised manuscript.
Reviewer 2 Report
This manuscript investigates the relationship between the syntactic accessibility levels of street networks that make up the urban fabric and the numbers of commercial uses and food services within these regions. Although this is not a new research, it has some interesting findings. However, the manuscript needs to be revised fully before being publishable.
Below are my comments:
- Abstract:
There are many sentences/concepts that need to be clarified in the Abstract so that the reader understands fully the scope and findings of this study.
The first sentence claims that the relationship between urban morphology and urban function distribution is crucial for sustainability without giving any reason why. This should be clarified and substantiated.
Line 13: “quantified the correlation”- correlation is already a quantification, so this sentence is wrong.
Line. 14: “urban information”- what does this mean? Needs to be clarified.
Line 21: do authors mean "the potential influence of street network layout on the distribution of businesses?" This also should be clarified.
Line 28: “ the strongest level of urban spatial centrality”- this is not clear, needs to be clarified.
- Introduction:
The Intro is lacking any summary/description of previous studies and their findings regarding the subject. This is important to lay the base of the significance of the topic and identify the missing gaps in literature that this study intends to fill. Either a separate Lit Review section should be included, or these studies should be summarized under Introduction.
Line 54: “spatial subjectivity”- not sure of this-what exactly "spatial subjectivity" means? Space syntax is a discreet and objective method, so why subjectivity?
Line 77-78: what is the advantage of using mathematical models and Big Data? need to explain.
Lines 82-89: This is about the Methodology-so should not be stated here. Instead, the authors should state why this study is needed, what are the aims of the study and how it is contributing to the existing knowledge.
- Methodology:
Lines 96-97: complex and diverse in terms of what? land use types, street types, etc??
Lines 98-99: “caused challenges in terms of integrity and rationality”- need to clarify, such as what?
Line 100: "Edge effect" should be defined for readers outside of the space syntax circle..
The authors need to detail the size of the study area- i.e., X by Y square meters...
Line 112: what are line segments? do you mean road segment, i.e. the segment between each intersection? need to clarify.
Figure 2: not legible-please revise. maybe try using white background?
Line 117: what information do POI data include? parcel-based land-uses? just floor areas or all total areas? Please explain. Also need to explain why only food service and shopping were selected as commercial land uses.
Line 132: what does "aggregation effect" mean? you mean centrality?
Table 1: The formula for Choice is wrong: the one stated by the authors define “Connectivity”. Please see the correct formula in the paper below:
In: Li, Xin, Zhihan Lv, Zhigao Zheng, Chen Zhong, Ihab Hamzi Hijazi, and Shidan Cheng. “Assessment of lively street network based on geographic information system and space syntax.” Multimedia Tools and Applications 76, no. 17 (2017): 17801-17819.
Line 139-148: This paragraph does not relate to syntactic analysis, so needs to be moved to the below paragraph.
Line 153: This claim is not correct- space syntax analyses also include angular and directional analysis too, so why did the authors use metric analysis only and not run stepwise analysis i.e. r:3, r:5, etc.?
Line 164: Table 2 refers to 15 metric radii (including n), not 16?
Line 166: “16 times” – do the authors mean "Depthmap was used to calculate analysis at 16 different scales"?
Line 174: Figure 6b shows that the grid is divided based on urban blocks and not streets; so maybe this sentence needs to identify this.
Lines 184-186: “we have used different statistical methods in different research units”- I think authors used different “units of measurements” and not “different stats methods” and also why? what is the point of correlation with the max and min values- why not only study mean values?
Table 3: is this the log value of total number of POIs?
Lines 226: “positioning of the commercial layout was relatively blurry”- needs to be clarified, not clear.
Figure 8: “number of POI” is missing in this Figure.
Figure 9: what is the number of data points in this correlation?-same for the other correlational analyses? The n (data points) should be noted in under these analyses.
Line 287-288: “the correlation decreases as the radius increases” - negative correlation means: the number of establishment decreases as the syntactic values of areas increase..Similarly, Line 313: positive correlation means as the syntactic measures increased, number of POIs increased. The strengthening of correlation in parallel to units of scale is another phenomena.
Line 311: “Fig.9” should be “Fig.11”
Line 330: “Fig.10” should be “Fig.12”
Lines 354-356: “Metric mean depth” should be introduced in the section of space syntax measures; otherwise, it is confusing. This sentence does not make sense at all- max metric mean depth is close to 20km, but min metric mean depth is much smaller than 10km? Needs clarification what this means and how it sheds light on the close relationship btw variables at 10-20 km scales.
- Discussion:
Line 361: What does “positive urban morphology” mean? Needs clarification.
Lines 368-369: not so sure about this- why not conduct regression analyses- putting all syntactic measures together and seeing the comparative roles of each in relating to the land uses? Also, why create a grid- and block-based spaces to analyse this relationship- why not do a regression with the number of establishments on each road segment and see whether the syntactic value of each segment is related to the number of establishments on it? like:
Scoppa, Martin D., and John Peponis. "Distributed attraction: the effects of street network connectivity upon the distribution of retail frontage in the City of Buenos Aires." Environment and Planning B: Planning and Design 42, no. 2 (2015): 354-378.
Line 433: “through different statistical methods” – as stated above; not clear-you have used t-test to measure the relationship; what are the other statistical methods??
Line 467: what does "equalize urban space" mean?
Most importantly, what are the Implications of these findings? How does it contribute to literature and planning? This is a very important part that needs to be highlighted.
- The language should be revised to correct all the grammar problems. For example:
Line 21: “changes within changes in scale”; line 47: needs to be re-written as “for studying urban form are necessary.”; line 134: “potential space” should be “accessibility potential of a space”?
Author Response
General response:
We greatly thank the four anonymous Reviewers for putting forward the constructive comments essential for improving the quality of our paper. In this version of revision, we substantially revised the Abstract, Introduction, and Discussion by considering all reviewers’ suggestions, and especially addressed the contributions of our study in these parts. We also added some details to the method in order to make it clearer. We reorganized the words of results to better express our findings. We provide two versions of the revised manuscript, one is in track change and one is in no track. For smooth reading, we highly recommend the no track version, as the line number mentioned below are referred to the line position in the no track version. Finally, the whole paper was extensively polished for its language. Below are the specific responses to individual comments.
Response to Reviewer 2
This manuscript investigates the relationship between the syntactic accessibility levels of street networks that make up the urban fabric and the numbers of commercial uses and food services within these regions. Although this is not a new research, it has some interesting findings. However, the manuscript needs to be revised fully before being publishable. Below are my comments:
Q1. Abstract: There are many sentences/concepts that need to be clarified in the Abstract so that the reader understands fully the scope and findings of this study. The first sentence claims that the relationship between urban morphology and urban function distribution is crucial for sustainability without giving any reason why. This should be clarified and substantiated.
A: Thank you for your advice. We made an extensive modification to provide the research background of our study, as well as the current research gap and the potential findings. In this way, it is easier for readers to understand. Please see the Introduction in the revised manuscript.
Q2. Line 13: “quantified the correlation”- correlation is already a quantification, so this sentence is wrong.
A: Thanks for pointing this out for us. We changed it into ‘quantification of the relationship’ please see Line 11-13 in the revised manuscript.
Q3. Line. 14: “urban information”- what does this mean? Needs to be clarified.
A: Sorry for the ambiguity. In the article, the ‘urban information’ is mainly referred to the information of POI data (shopping and food service) in the city. We rewrote these sentences.
Q4. Line 21: do authors mean "the potential influence of street network layout on the distribution of businesses?" This also should be clarified.
A: Yes, thank you for your reminder. We have already clarified. Please see the Abstract in the revised manuscript.
Q5. Line 28: “ the strongest level of urban spatial centrality”- this is not clear, needs to be clarified.
A: Sorry for the unclearness. We rewrote the abstract to make our findings more clear. Please see the Abstract in the revised manuscript.
Q6. Introduction: The Intro is lacking any summary/description of previous studies and their findings regarding the subject. This is important to lay the base of the significance of the topic and identify the missing gaps in the literature that this study intends to fill. Either a separate Lit Review section should be included, or these studies should be summarized under Introduction.
A: Thanks for your suggestion. We rewrote the Introduction to (1) point out the scope of our study, (2) summarize the research gap of current studies, (3) state the objective of our study, and (4) the potential findings of our research. Please see the Introduction in the revised manuscript.
Q7. Line 54: “spatial subjectivity”- not sure of this-what exactly what "spatial subjectivity" means? Space syntax is a discreet and objective method, so why subjectivity?
A: Sorry for the misunderstanding here. The "space subjectivity" here is actually the space ontology. We hope to emphasize the influence of urban space on the city through space syntax.
Q8. Line 77-78: what is the advantage of using mathematical models and Big Data? need to explain.
A: Done. Please see Line 68-74 in the revised manuscript.
Q9. Lines 82-89: This is about the Methodology-so should not be stated here. Instead, the authors should state why this study is needed, what are the aims of the study and how it is contributing to the existing knowledge.
A: Thanks. We followed your advice and rewrite this part. Please see the Introduction in the revised manuscript.
Q10. Methodology: Lines 96-97: complex and diverse in terms of what? land use types, street types, etc??
A: Sorry for the unclearness. We revised it to be more specific. Please see Line 111-115 in the revised manuscript.
Q11. Lines 98-99: “caused challenges in terms of integrity and rationality”- need to clarify, such as what?
A: Sorry for the difficulty in understanding. Xi'an, as one of the famous historical capitals, has certain representativeness in the study of complex and diverse urban forms. However, due to this morphological feature, how to clarify the complex form of the city without destroying the integrity of the city's space, so that the rational planning of urban functions becomes the focus of research. For example, what kind of connection and planning should be carried out between the space of the old city and the new development area to preserve the original spatial texture of the old city and make reasonable adjustments to the problem area. Or how to maintain the connection between the old city and the new district to maintain the integrity of the entire area and avoid excessive spatial isolation. These are the challenges faced by urban with complex urban morphology.
Q12. Line 100: "Edge effect" should be defined for readers outside of the space syntax circle. The authors need to detail the size of the study area- i.e., X by Y square meters...
A: Done. We add the explanation of Edge effect and adding the cover area of the study area. Please see Line 116-123 in the revised manuscript.
Q13. Line 112: what are line segments? do you mean road segment, i.e. the segment between each intersection? need to clarify.
A: Each line segment is the piece of street between two street junctions. We added this in the paper.
Q14. Figure 2: not legible-please revise. maybe try using white background?
A: I'm sorry because Figure 2 was automatically intercepted by the software—Baidu Maps' Interceptor. There is only a black background. And there is no way to convert it to white. But we have processed the white road to improve the display of white. We expect it should have improved.
Q15. Line 117: what information do POI data include? parcel-based land-uses? just floor areas or all total areas? Please explain. Also, need to explain why only food service and shopping were selected as commercial land uses.
A: In fact, without considering the time and economic costs, commercial POI data can contain a variety of data information. For example, the name of the business, geographic location, building area, per capita consumption level, business type, etc. Among them, this research mainly focuses on the distribution of business. Therefore, we are mainly involved in the geographic locations of commercial POI data. Besides, shopping and food service are the most basic urban commercial functions. At the same time, they are closely related to people's life. However, we can only access two kinds of commercial POIs (i.e. food services and shopping) for our study. We also explained this in the revised paper.
Q16. Line 132: what does "aggregation effect" mean? you mean centrality?
A: I am sorry that this is not clearly expressed. The "aggregation effect" here refers to centrality. We have revised it accordingly.
Q17. Table 1: The formula for Choice is wrong: the one stated by the authors define “Connectivity”. Please see the correct formula in the paper below:
In: Li, Xin, Zhihan Lv, Zhigao Zheng, Chen Zhong, Ihab Hamzi Hijazi, and Shidan Cheng. “Assessment of lively street network based on geographic information system and space syntax.” Multimedia Tools and Applications 76, no. 17 (2017): 17801-17819.
A: Thank you for pointing this out for us. We have corrected the error based on the reference [2] you provided. Please see Table 1 in the revised manuscript.
Q18. Line 139-148: This paragraph does not relate to syntactic analysis, so needs to be moved to the below paragraph.
A: Done. We moved it to Section 2.5.
Q19. Line 153: This claim is not correct- space syntax analyses also include angular and directional analysis too, so why did the authors use metric analysis only and not run stepwise analysis i.e. r:3, r:5, etc.?
A: Thank you for your advice. The measured radius of space syntax includes topological radius and metric radius. However, urban with more uniform morphology and structure requires more de-tailed expression [3] such as the grid-like streets of Xi'an. This expression is the segment expression based on the axis model and road centerline data [4]. Among them, the angle analysis in segment analysis is considered to be very consistent with human behavior [5]. At the same time, the angle analysis under the limit of the metric distance radius is helpful to extract the main reaching traffic and traversing traffic paths in the urban street network [6]. Therefore, this research is more inclined to use the metric distance radius to establish different research scales. We have also revised this statement. Please see Line 163-168 in the revised manuscript.
Q20. Line 164: Table 2 refers to 15 metric radii (including n), not 16?
A: Sorry for forgetting to enter "35 km" in the large scale group. It has now been added.
Q21. Line 166: “16 times” – do the authors mean "Depthmap was used to calculate analysis at 16 different scales"?
A: Thank you for your advice. That’s right. "16 times" mean we used Depthmap to calculate analysis at 16 different scales that are exactly as listed in Table 2.
Q22. Line 174: Figure 6b shows that the grid is divided based on urban blocks and not streets; so maybe this sentence needs to identify this.
A: Done.
Q23. Lines 184-186: “We have used different statistical methods in different research units”- I think authors used different “units of measurements” and not “different stats methods” and also why? what is the point of correlation with the max and min values- why not only study mean values?
A: Sorry for the ambiguity. Actually, we try to express we used different statistic metrics, i.e. mean, max and total. We revised all the related parts in the paper.
Q24. Table 3: is this the log value of total number of POIs?
A: Yes, we revised it. Please see Table 3 in the revised manuscript.
Q25. Lines 226: “positioning of the commercial layout was relatively blurry”- needs to be clarified, not clear.
A: Sorry, actually we want to say the contrast of density becomes less prominent under the block units. We modified the expression in the revised manuscript.
Q26. Figure 8: “number of POI” is missing in this Figure.
A: Sorry for not elaborate on it. Since the cover area of each block is different from each other. Larger blocks tend to contain more POIs. Therefore, it is difficult to map. Thus we divided the number of POIs within a block with its cover area and denote it as P'. The calculation formula is:
Where Ni represents the number of POIs within block i and Si represents the cover area of block i.
We have also explained in the article. Please see Figure 8.
Q27. Figure 9: what is the number of data points in this correlation?-same for the other correlational analyses? The n (data points) should be noted under these analyses.
A: Thank you for your advice. In this article, the number of grid units is 5239 and the number of block units is 1559. Therefore, when performing correlation analysis based on grids, n=5239 for all cases. When performing correlation analysis based on blocks, n=1559 for all cases. We added this information in the corresponding figure captions.
Q28. Line 287-288: “the correlation decreases as the radius increases” - negative correlation means: the number of establishments decreases as the syntactic values of areas increase. Similarly, Line 313: positive correlation means as the syntactic measures increased, the number of POIs increased. The strengthening of correlation in parallel to units of scale is another phenomenon.
A: Thanks for pointing this out for us. We have changed our presentation error. Please see Section 3.3.
Q29. Line 311: “Fig.9” should be “Fig.11”
A: Done.
Q30. Line 330: “Fig.10” should be “Fig.12”
A: Done.
Q31. Lines 354-356: “Metric mean depth” should be introduced in the section of space syntax measures; otherwise, it is confusing. This sentence does not make sense at all- max metric mean depth is close to 20km, but min metric mean depth is much smaller than 10km? Needs clarification on what this means and how it sheds light on the close relationship btw variables at 10-20 km scales.
A: Thank you for your advice. Because the Metric mean depth has a certain connection with the urban scale. Therefore, in the article, we made a relatively simple comparison between the scales and the Metric mean depth and proposed corresponding assumptions. Is there a certain connection between the two of them or is it just a coincidence that they are numerically similar? This requires us to further study in the next research. Therefore, we did not go into too much detail. In theory, the Metric mean depth expresses the degree of integration between the local space and the overall space of the city. The research scale also reflects the relationship between local space and overall space. Therefore, this hypothesis has a certain theoretical basis. However, no clearer explanation has been found so far.
Q32. Discussion: Line 361: What does “positive urban morphology” mean? Needs clarification.
A: Sorry for the ambiguity. We changed it to ‘sustainable urban morphology’.
Q33. Lines 368-369: not so sure about this- why not conduct regression analyses- putting all syntactic measures together and seeing the comparative roles of each in relating to the land uses? Also, why create grid- and block-based spaces to analyze this relationship- why not do a regression with the number of establishments on each road segment and see whether the syntactic value of each segment is related to the number of establishments on it? like: Scoppa, Martin D., and John Peponis. "Distributed attraction: the effects of street network connectivity upon the distribution of retail frontage in the City of Buenos Aires." Environment and Planning B: Planning and Design 42, no. 2 (2015): 354-378.
A: Thank you for your advice. As you mentioned in the article[7], the author mainly studied the impact of each road on the distribution of surrounding businesses. This to a certain extent belongs to the study of linear space. In our research, we turn the research object into a area research. This can be more conducive to further research on urban land use. At the same time, this also provides a new research method for land use evaluation. But when it comes to the study of area data such as land use, the division of space needs to be considered. Therefore, on the one hand, we divide the research area into equal space based on the grid level, and on the other hand, divide the urban space according to the block according to the urban texture.
Q34. Line 433: “through different statistical methods” – as stated above; not clear-you have used t-test to measure the relationship; what are the other statistical methods??
A: Sorry for the ambiguity. Actually, we try to express we used different statistic metrics, i.e. mean, max and total. We revised all the related parts in the paper.
Q35. Line 467: what does "equalize urban space" mean?
A: Thank you for your advice. I'm sorry that there is no clear statement here. What we try to express is that considering the rather limited cover area of 500 m X 500 m grids, the space syntax variables within each grid will not change greatly, because the length of line segments generally greater than 500 m.
Q36. Most importantly, what are the implications of these findings? How does it contribute to literature and planning? This is a very important part that needs to be highlighted.
A: Thank you for your advice. The implications of our finding are specifically addressed in the Discusssion, Conclusion, and Abstract. Please see the revised manuscript.
Q37. The language should be revised to correct all the grammar problems. For example:
Line 21: “changes within changes in scale”; line 47: needs to be re-written as “for studying urban form are necessary.”; line 134: “potential space” should be “accessibility potential of a space”?
A: Thanks. The writing was extensively polished in this version to make it scientifically sound and much more readable.
References:
- Duan, J.; Hillier, B. Space Syntax and Urban Planning; Southeast University Press: 2007.
- Li, X.; Lv, Z.; Zheng, Z.; Zhong, C.; Hijazi, I.H.; Cheng, S. Assessment of lively street network based on geographic information system and space syntax. MULTIMED TOOLS APPL 2015, 17801-17819.
- Dalton, N.S., Fractional configuration analysis and a solution to the Manhattan problem. In Proceedings of the 3rd International Symposium on Space Syntax, Georgia Atlanta:Georgia Institute of Technology, 2001.
- Dalton, N.S.; Peponis, J.; Dalton, R., To tame a TIGER one has to know its nature: extending weighted angular integration analysis to the description of GIS road-centerline data for large scale urban analysis. In Proceedings of the 4th International Space Syntax Symposium, London:University College London, 2003.
- Hillier, B.; Iida, S., Network and psychological effects in urban movement. In Proceedings of Spatial Information Theory, Berlin:SpringerVerlag, 2005.
- Turner, A., Angular analysis. In Proceedings of the 3nd International Symposium on Space Syntax, 2001.
- Scoppa, M.D.J.P. Distributed attraction: the effects of street network connectivity upon the distribution of retail frontage in the City of Buenos Aires. Environment and Planning B: Planning and Design 2015, 42, 354-378.
Reviewer 3 Report
The article presents an interesting research conducted in a well elaborated manner. It would be advisable to add a segment (within the discussion part) which would clarify the applicative potential/possibilities of the research/selected methodology for both urban planning and urban design since this is not clearly explained.
Author Response
General response:
We greatly thank the four anonymous Reviewers for putting forward the constructive comments essential for improving the quality of our paper. In this version of revision, we substantially revised the Abstract, Introduction, and Discussion by considering all reviewers’ suggestions, and especially addressed the contributions of our study in these parts. We also added some details to the method in order to make it clearer. We reorganized the words of results to better express our findings. We provide two versions of the revised manuscript, one is in track change and one is in no track. For smooth reading, we highly recommend the no track version, as the line number mentioned below are referred to the line position in the no track version. Finally, the whole paper was extensively polished for its language. Below are the specific responses to individual comments.
Response to Reviewer 3
Q1: The article presents an interesting research conducted in a well elaborated manner. It would be advisable to add a segment (within the discussion part) which would clarify the applicative potential/possibilities of the research/selected methodology for both urban planning and urban design since this is not clearly explained.
A: Thank you for your advice. We purposely address the methodological and practical implications of our study. Please Discussion and Conclusion in the revised manuscript.
Reviewer 4 Report
The article under review examines the scale factor in a comparative study of urban morphology and the location of retail outlets in the city of Xi’an (China). The paper is written in clear language, is well structured, contains a thorough discussion of the methodology involved in the study and a detailed description of the results obtained. The central question of the study, the role of the scale factor in the study of the urban environment through syntactic analysis, has a considerable research potential. The work is well illustrated. The references are well chosen.
The work can be published as presented.
Author Response
General response:
We greatly thank the four anonymous Reviewers for putting forward the constructive comments essential for improving the quality of our paper. In this version of revision, we substantially revised the Abstract, Introduction, and Discussion by considering all reviewers’ suggestions, and especially addressed the contributions of our study in these parts. We also added some details to the method in order to make it clearer. We reorganized the words of results to better express our findings. We provide two versions of the revised manuscript, one is in track change and one is in no track. For smooth reading, we highly recommend the no track version, as the line number mentioned below are referred to the line position in the no track version. Finally, the whole paper was extensively polished for its language. Below are the specific responses to individual comments.
Response to Reviewer 4
Q1. The article under review examines the scale factor in a comparative study of urban morphology and the location of retail outlets in the city of Xi’an (China). The paper is written in clear language, is well structured, contains a thorough discussion of the methodology involved in the study and a detailed description of the results obtained. The central question of the study, the role of the scale factor in the study of the urban environment through syntactic analysis, has considerable research potential. The work is well illustrated. The references are well chosen. The work can be published as presented.
A: Thanks for your appreciation of our study.
Round 2
Reviewer 1 Report
All my concerns have been well addressed - I enjoy reading the improved work.
Reviewer 2 Report
I would like to thank the authors for their extensive editing. I think the paper is much more clear and is now publishable, but I would urge the authors/editors to have a final English check as there are still some grammatical mistakes.